# Atomic Force Microscopy (AFM) Applications in Arrhythmogenic Cardiomyopathy

**DOI:** 10.3390/ijms23073700

**Published:** 2022-03-28

**Authors:** Brisa Peña, Mostafa Adbel-Hafiz, Maria Cavasin, Luisa Mestroni, Orfeo Sbaizero

**Affiliations:** 1CU-Cardiovascular Institute, University of Colorado Anschutz Medical Campus, 12700 East 19th Ave., Aurora, CO 80045, USA; brisa.penacastellanos@cuanschutz.edu (B.P.); luisa.mestroni@cuanschutz.edu (L.M.); 2Department of Bioengineering, University of Colorado Anschutz Medical Campus, 12700 East 19th Ave., Aurora, CO 80045, USA; mostafa.abdel-hafiz@cuanschutz.edu; 3Department of Medicine, Division of Cardiology, University of Colorado Anschutz Medical Campus, 12700 East 19th Ave., Aurora, CO 80045, USA; maria.cavasin@cuanschutz.edu; 4Consortium for Fibrosis Research & Translation, University of Colorado Anschutz Medical Campus, 12700 East 19th Ave., Aurora, CO 80045, USA; 5Department of Engineering and Architecture, University of Trieste, Via Valerio 10, 34127 Trieste, Italy

**Keywords:** atomic force microscopy, arrhythmogenic cardiomyopathy, biomechanics, tissue stiffness, fibrosis

## Abstract

Arrhythmogenic cardiomyopathy (ACM) is an inherited heart muscle disorder characterized by progressive replacement of cardiomyocytes by fibrofatty tissue, ventricular dilatation, cardiac dysfunction, arrhythmias, and sudden cardiac death. Interest in molecular biomechanics for these disorders is constantly growing. Atomic force microscopy (AFM) is a well-established technic to study the mechanobiology of biological samples under physiological and pathological conditions at the cellular scale. However, a review which described all the different data that can be obtained using the AFM (cell elasticity, adhesion behavior, viscoelasticity, beating force, and frequency) is still missing. In this review, we will discuss several techniques that highlight the potential of AFM to be used as a tool for assessing the biomechanics involved in ACM. Indeed, analysis of genetically mutated cells with AFM reveal abnormalities of the cytoskeleton, cell membrane structures, and defects of contractility. The higher the Young’s modulus, the stiffer the cell, and it is well known that abnormal tissue stiffness is symptomatic of a range of diseases. The cell beating force and frequency provide information during the depolarization and repolarization phases, complementary to cell electrophysiology (calcium imaging, MEA, patch clamp). In addition, original data is also presented to emphasize the unique potential of AFM as a tool to assess fibrosis in cardiac tissue.

## 1. Introduction

Arrhythmogenic cardiomyopathy (ACM) is a heart disease that affects the myocardium and causes arrhythmias and sudden death. Frequently, it manifests during puberty in otherwise healthy young individuals and is a known cause of sudden death in young athletes. Indeed, exercise has been documented to accelerate its progression [1,2,3,4]. ACM is often a genetic condition caused by a ‘mutated’ gene, resulting in impaired mechanical properties of cardiac cells. ACM is commonly referred to as a disease of the desmosome [5,6,7]. However, outside the desmosome, additional ACM-linked genes have been associated with this disease including genes related (i) to the transmembrane protein 43 (TMEM43) and N-cadherin (CDH2), (ii) to the cell nucleus (LMNA), (iii) to the cytoskeleton (desmin (DES), filamin (FLNA) and α-T-catenin (CTNNA3), (iv) to the tight junction protein-1 (TJP1) and (v) to the voltage-gated sodium channel alpha subunit 5 (SCN5A) [8,9,10,11,12,13,14,15,16,17]. This discovery drove [18] many studies designed to understand the relationship between genes abnormalities and ACM pathological and clinical findings [18], since the genetic cause, subtype, and incomplete penetrance results in significant variability in disease characteristics between individuals [19]. It is therefore clear that the genetics of AC is still complex and not fully understood. In fact, it is acknowledged that AC phenotypes are activated and boosted by strenuous endurance physical activity when excessive mechanical stresses and repetitive adrenergic stimulation are present. Understanding the arrhythmia mechanisms in the early stage of the disease is therefore vital to reduce the risk of, and potentially prevent, lethal arrhythmias in young people. Despite mechanical stresses having been proved to be so important in this disease, since both cardiac function and mechanical integrity strictly depend on correct cell–cell and cell–extracellular matrix (ECM) connections, it remains puzzling to replicate the heart mechanical load during physical activity in laboratories. Desmosomes, together with integrins and cadherins, are protein complexes that connect adjoining cells and the extracellular matrix, and therefore are highly expressed in tissues subject to mechanical stresses, such as heart. Furthermore, desmosomes are coupled to structural mechanoresponsive cytoskeletal elements, such as actin, microtubules, and intermediate filaments, which allow cardiomyocytes (CMs) to adapt to external and internal stresses. These mechano-sensible elements are essential for accurate application of mechanical loads and for efficient mechanotransduction. Mechanotransduction is therefore defined as the mechanism by which force transmission between ECM–cell and between the cell itself is translated into a series of intracellular signaling events [20]. However, the choice of the most suitable technique to be used to assess mechano-transduction is not trivial since cell response to the mechanical forces exerted by the cell microenvironment, including neighboring cells and the extracellular matrix, can be elastic, viscous, or viscoelastic. Recent findings in cell mechanics have shown that changes in cell and nuclear mechanics occur in many human diseases: cancer, several cardiovascular diseases, microbe interactions in infectious diseases, and frailty in aging [21,22,23]. The molecular mechanisms of cellular mechano-transduction have been studied for almost 40 years, but due to the complexity of the cellular biomechanics, a single cell experiment which isolates the cell from its environment does not accurately mimic the in vivo conditions. However, in recent years, new tools such as high-resolution microscopy, atomic force microscopy, magnetic twisting cytometry, particle-tracking micro-rheology, parallel-plate rheometer, cell monolayer rheology, and optical stretching, optical tweezers, 3-D cell culture, loading mechanisms, gene chip analysis, and computer modeling have provided additional aid into discovery of the mechanisms involved in mechanotransduction. The goal of this review is to discuss the Atomic Force Microscopy (AFM) applications in ACM and describe the possible results to address the aforementioned cell mechanical questions. A word of caution before proceeding: values of cell mechanical properties determined by AFM and found in the literature differ substantially, even though groups use the same instrument. Reasons for these deviations are due to the physical and physiological cues of the studies. The geometry of the probe tips, the indenting force, the AFM mode, the AFM analysis methodology, and the differences in cell-culture conditions (cell passage number, pH, temperature, etc.) significantly influence the nanomechanical properties of biological samples. Indeed, sharp probe tips create a two-fold increase in the value of the effective Young’s modulus relative to that of the blunt tips (2, 3). Therefore, it is very important that the physical and physiological cues are kept constant during the analysis in order to identify biomechanical differences between samples using AFM. Although this reduces the possibility of comparing different datasets, the AFM is a powerful technique for assessing cell mechanics and can even lead to the use of some of these instruments for clinical applications.

## 2. Cell Mechanics and Mechanobiology

The observation of physical and mechanical phenomena in living organisms is not a recent discovery. Wilhelm His proposed the idea that cells are subjected to the laws of physics in 19th century, when Julius Wolff also postulated his famous law on bone remodeling [24,25].

Physics and mechanics were thought to be related to life, but any effort in this field was unfortunately torn apart. Proper technologies that could have tested and confirmed any hypotheses were lacking. The scientific interest was therefore focused on unraveling molecular and genetic mechanisms.

The advent of sophisticated tools and nanotechnologies again called the attention of academics to the role played by physical events in biology. A new era then started, mechanobiology and biomechanics showed up as a rapidly growing field of study.

Mechanobiology is the discipline that studies the effects induced by physical cues on living systems, at any scale: hence, it also studies the mechanotransduction. Mechanobiology is often overlapped with biomechanics, but there might be distinct differences. Biomechanics is the study of structural and mechanical properties of a living system, and when the living system under investigation is a cell, it is termed cell mechanics. That begs the questions: where should we focus our efforts? Cell mechanics or mechanobiology? The answer is quite simple, since the two are mutually dependent. Indeed, cell mechanical properties are crucial for mechanotransduction mechanisms, and mechanotransduction can trigger changes in the cell mechanical properties. Notably, experimental procedures are usually directed to either mechanobiology or cell mechanics; ergo, they need to be integrated with other techniques to entirely describe a phenomenon.

Let us describe just two examples of mechanobiology and cell mechanics, trying to stimulate the interest of the readers in these fields and convince them about the importance of this research.

One of the hottest trends that drove science in the last few years is certainly the study of stem cells, since they can differentiate between different cell types and therefore be used in regenerative/reparative medicine. Researchers have discovered that mechanical aspects cannot be ignored when dealing with stem cells. Indeed, these cells are largely influenced by the stiffness of the surrounding environment, namely cells exhibit different behavior as they sense either a soft or stiff substrate [26]. This behavior has been demonstrated using human mesenchymal stem cells (MSCs) which respond to substrate stiffness, differentiating into various lineages: for example soft matrices lead to neurogenic, while stiffer matrices lead to either myogenic or osteogenic [26]. Osteogenic differentiation has been observed also on stiff substrates in human periodontal ligament stem cells (PDLSCs) [27]. Conversely, muscle stem cells (MuSCs) retained their “stemness” and their regenerative potential when cultured on compliant substrates [28].

Another relevant theme that captivates major efforts in the scientific community is cancer, which is the second leading cause of death worldwide, and has a huge economic impact (around 1.16 trillion USD in 2010). Despite the progress made so far, cancer remains an issue. Biomechanics is playing a pivotal role in revealing the mysteries of cancer. In particular, cell mechanics techniques showed that cancer cells, especially the highly invasive ones, are more deformable than controls [29]. This is potentially important to explain the mechanisms of cancer progression, since metastatic cells must be very deformable to intra- and extravasate, survive the blood stream, and invade other sites [30]. This information needs to be complemented by other experiments (e.g., directed to the molecular mechanisms). Nonetheless, these results could eventually lead to identifying mechanical features as therapeutic targets, which is one of the key goals for researchers in the field.

After this introductive section, the focus is now pointed to atomic force microscopy (AFM), one of the techniques that contributed the most to the success of cell mechanics.

## 3. AFM—The Mechanical Machine

AFM was first described in 1986 [31], and it is now a well-established technique that can be performed under physiological conditions, and thus, it can be used for imaging and biomechanics characterization of biological samples at the nanoscale level [32]. The AFM is a scanning probe microscope in which a physical probe indents, scans, and images a specimen. It consists of a cantilever that holds a fixed probe; the probe which can have a tip or not, depending on its geometry, is the one that indents or scan the specimen. A laser beam pointing on the cantilever surface can be reflected and collected by a matrix of photodiodes. Whenever the probe interacts with a sample, the cantilever bends, and the deflection is detected as a variation of the currents transduced by the photodiodes.

The characteristics of the cantilever assembly, such as material and probe tip shape, strongly depend upon the application. Silicon (Si) and silicon nitride (Si_3_N_4_) are a common choice [33]. Silicon nitride cantilevers are often coated with gold in order to enhance their reflectivity as well as to facilitate cantilever-tip functionalization (gold allows exploiting the thiol-gold chem), which is required for some chemical and biological applications. Unfortunately, the gold coating promotes sensitivity to small temperature drifts during analysis [33].

Cantilevers systems (probe and probe tips) can be made out of distinct materials. For instance, CP- PNP-SiO probes (NanoAndMore Gmbh) have a Si_3_N_4_ cantilever, but the spherical probe is made of silicon dioxide (SiO_2_). The probe, the probe tip and the cantilever may have various shape and geometries (e.g., sphere or pyramid for the probe, cone for the tip, triangle, or rectangle for the cantilever, etc.).

AFM probe customizations are endless, but particular consideration may be given to the mode in which the AFM will operate, the environment, and the type of specimen. The following sections provide detailed information regarding specific techniques using AFM for biological applications with two main AFM modes: (i) force spectroscopy and (ii) scanning methodologies; the potential to use these two modes to assess ACM disease will be further discussed in the final sections of this review.

## 4. Force Spectroscopy with AFM

### 4.1. Single Molecule Force Spectroscopy

Single-molecule force spectroscopy (AFM-SMFS) is a procedure that studies the mechanical properties of single molecules. It is a terrific tool for understanding protein–protein interactions, protein folding kinetics, and protein unfolding pathways [34]. In this case, the AFM cantilever is above the sample which contain the studied proteins immersed in a buffer. The test is initiated by getting together the cantilever and the sample (protein). When the protein and the AFM tip are in contact, the AFM tip starts moving up away from the surface, pulling on the adsorbed protein. The pulling force that is exerted by the protein on the cantilever is determined by Hooke’s law through the deflection of itself. In a typical experiment, the protein of interest would give a characteristic saw-tooth pattern (Figure 1) where each “rip” corresponds to the unfolding of one of the domains in the polyprotein. The magnitude of force generated with the AFM is sufficient to unfold single proteins and nucleic acid structures. Data are usually analyzed using the worm-like chain model which approximates the molecular extension of a semiflexible chain (like a protein). By fitting each peak in the force–extension curve (Figure 1), data such as contour length and persistence length can be deduced. The contour length of a polymer chain is the physical maximum extension that the latter can achieve. On the other hand, persistence length is a basic mechanical property that quantifies a polymer’s stiffness.

### 4.2. Single Cell Force Spectroscopy

In AFM single cell force spectroscopy, the aim is to obtain a plot of the force versus the indentation depth which provides mechanical data regarding a cell [35]. The approach (AFM tip loading the cell) part of the curves provides the cell elasticity (Figure 2). The slope of the curve is related to the resistance of the cell to the applied force, therefore, the lower the slope, the higher the elasticity. The elasticity data achieved from such analysis is considered as the cell Young’s modulus. This assumption is only valid if the cell performs in an elastic manner. In the case of cells, it is a realistic assumption if the experimental conditions are correctly chosen. At the beginning of a force–distance experiment, the cantilever-tip assembly is not in contact with the sample. The probe is moved toward the specimen until the contact occurs, the tip exerts a certain force, and penetrates/indents the sample; then, the tip and cantilever are retracted from the sample until the contact is lost. The contact period is termed dwell time (Figure 2) [36].

These experiments usually result in the plot of cantilever deflection against the movement of the piezoelectric scanner (i.e., a distance). These plots are commonly known as force–distance (F-d) curves. The mechanical properties of a sample are inferred from F-d curves. Generally, if the dwell time is null, the user can measure both stiffness (usually in terms of Young’s modulus) and adhesive properties. Otherwise, if the dwell time is not null, stress relaxation and creep compliance can be assessed. Performing F-d experiments on single cells can be challenging, and several precautions must be considered. For instance, if the cantilever is stiff, the deflection might be difficult to detect, whereas if it is too soft, thermal vibrations can interfere, and thus, the overlapping of the F-d curve or the indentation might not be adequate to reliably estimate mechanical properties [37]. When evaluating cell stiffness, consideration may want to be given to the following.

(i) The intrinsic viscoelastic behavior of a cell must always be taken into account. The tip has to be moved slowly against the sample (low indentation speed) in order to reduce the contribution of the hydrodynamic/viscous properties of the cell [38]. A typical range of indentation speeds is 50 nm/s ÷ 10 μm/s [32]. (ii) Living cells adhered to a substrate like petri dish or coverslip can be modelled as thin layers on hard substrates. The indentation must be within 10% of the cell height to lower the hard substrate contribute [37]. This represents a challenge when probing peripheral areas of the cell, where the sample thickness is low. (iii) The stiffness is commonly calculated in terms of Young’s modulus by fitting the F-d curves with an appropriate contact mechanics model. The most used is the Hertz model but it implies two main assumptions: the indenter must have a parabolic shape, and the sample under investigation is very much thicker than the indentation depth [39]. However, Sneddon extensions of the Hertz model take into account different indenter geometries. Other models like DMT (Derjagin, Muller, Toropov) and JKR (Johnson, Kendal, Roberts) are seldom used. (iv) When comparing the stiffness of identical cell types, the shape of the indenter is crucial. Indeed, it has been shown that cells appear softer when probed by spherical indenters as opposed to conical ones [40].

The important of assessing cell stiffness has been answered exhaustively by Martin Y.M. Chiang and his co-authors: “The impact of cell modulus (the deformability of cells or resistance to morphological change) extends beyond knowledge of a mechanical property to include cellular processes important in developmental biology, pathology, molecular biology, etc., as well as cell–material interactions in tissue engineering and regenerative medicine” [41].

AFM already proved to be very sensitive in discerning pathologic statuses. Back in 2005, Malgorzata Lekka and her group compared the deformability of erythrocytes from donors and hospitalized patients who were suffering from several disorders (coronary disease, hypertension, diabetes mellitus) [42]. Authors highlighted an increase in the Young’s modulus of erythrocytes derived from diabetes mellitus patients, as well as from cigarette smokers. Pathological erythrocytes were shown to be stiffer than healthy ones even in the context of anemias and anysocytosis [43]. The consequences of diabetes were also observed in cardiomyocytes from a murine model, finding out that the disease caused an increase of myocardium stiffness [44].

Since the scope of our review is to show the power of AFM in the study of genetic diseases, it is worth mentioning that the technique was already exploited. In this regard, studies on progeria in vitro models were remarkable, but they will be described in the dedicated section. Dilated cardiomyopathy (DCM) was previously investigated by our group. Indeed, Laurini et al. studied three LMNA mutations (E161K, D192G, N195K) known to cause DCM, discovering that these variants produce an increase of the Young’s modulus of neonatal rat cardiac myocytes [45]. A genetic cardiocutaneous disorder associated with a desmoplakin mutation has also been recently studied by our group with AFM, and keratinocytes derived from a patient skin biopsy exhibited a higher deformability (lower stiffness) than controls [46]. Moreover, AFM has also been used to prove that serum circulating proteins from pediatric patients with DCM cause cardiomyocyte stiffness [47].

### 4.3. Assessing the Cell Viscoelasticity

AFM force–deformation curves could also be used to evaluate cell viscoelastic behavior. If the cell response is elastic, the indentation and retraction curves should be identical, however, typically, there is a substantial dissimilarity between the “loading” and “un-loading” curve. This hysteresis cycle shows that the behavior is not purely elastic. Figure 3A [48] shows a typical time course of deflection signals during a test.

The AFM tip moves toward the cell surface with a speed of 1 μm/s (zone 1 in Figure 3A). The position where the force sharply increased characterizes the contact point between the tip and the cell membrane. After that, the cantilever base position is kept constant for a fixed time, while the cantilever’s force varies, and it is recorded with time (zone 2 in Figure 3A). Figure 3A shows that the loading force decays with time and this behavior is due to the cell viscoelasticity. After the loading time, the AFM tip is withdrawn (zone 3 in Figure 3A). Usually, a dwell period of 60s is used to reduce drift. The curves are normalized by setting the maximum force value to one, meaning all the relaxation data fell in the zero (minimum of the baseline) to one normalized force range. Force data recorded during the dwell time (relaxation phase) are then divided by the tip-cell contact area to get a stress data [48]. Experimental data might be fitted with a double-exponential Equation (2), corresponding to a Generalized Maxwell model with one spring in parallel with two Maxwell elements (Figure 3B). This model is usually chosen over single- and tri-exponential equations because the first does not well describe the first part of the relaxation curve, whereas the latter does not significantly enhance the quality of fitting.
(1)$$G\left(tot\right)=G_{0}+\sum_{i=1}^{2}G_{i}\;e^{-\frac{t}{\tau_{i}}}$$

In Equation (1), *G*(*tot*) is the normalized force profile and *τ*_1_ and *τ*_2_ are the two characteristic relaxation times. *G*_0_ corresponds to the Maxwell spring element and accounts for the equilibrium modulus, that is the asymptotic value of *G*(*tot*) for *t* → ∞. *G_i_* and *τ_i_* are the normalize force and relaxation time of the *i*-th Maxwell element, respectively. Considering Figure 3B, *τ_i_* is equal to *η_i_*/*G_i_*, where *η_i_* is the viscosity of the *i*-th Maxwell element. Specifically, *τ*_1_ governs the short-term relaxation behavior, whereas *τ*_2_ corresponds to the slow relaxation time. The normalized curve could also be used to calculate the percent relaxation at the end of the applied stress:(2)% relax=[1−E(t=60s)]×100

### 4.4. Assessing the Cell Adhesion Behavior

Cell adhesion is the process by which a cell binds to another entity, such as another cell, a substrate, or an organic matrix. Cell adhesion is fundamental in many processes, such as cell communication, tissue development, and preservation. It acts as a trigger for signals regulating cell differentiation, cell cycle, migration, and survival [49], and the interactions between cells and extracellular matrix (ECM) can influence cell behavior and function. Furthermore, cell adhesion plays a key role in health and disease: for example, the impairment of adhesiveness is typical in the pathogenesis of certain diseases, like ACM, where the speculated detachment of cardiomyocytes due to altered cell–cell junctions is thought to trigger the disorder. It has been previously reported by our group that neonal rat ventricula fibroblast (NRVF) carring LMNA D192G mutation present reduced cell–cell adhesion properties when compared with control (non-mutated NRVF [50]. Adhesive properties and events are usually derived from the retraction segment of a F-d curve. The overall cell adhesion may be described by the maximum adhesion force, the distance at which this force occurs, and the work of adhesion (Figure 4).

The maximum adhesion force, also called maximum detachment force, is represented by the peak of force with respect to the zero-force value. The distance at which this force occurs is calculated from the beginning of the withdrawal until the appearance of the peak. Finally, the work of adhesion, known as work of detachment, defines the energy dissipated during the retraction of the probe from the sample, and is calculated by integrating the area above the retract F- d curve and the zero-force value. “Single” bonds can be also inferred from F-d curves, looking at ruptures and tethers. The former are referred to cell adhesion receptors which are anchored to the cytoskeleton, whereas the latter involve receptors not linked to the cytoskeleton [33]. For the sake of completeness, it is worth noting that there are two main configurations to measure cell adhesion [33]:(1)a cell on a substrate is “touched” by either a tipless cantilever or a cantilever-tip assembly(2)a cell adhered to the cantilever is brought in contact with either a substrate or another cell.

In both cases, the substrate or the cantilever can be functionalized with proteins of interest (e.g., ECM proteins). In both configurations, specific adhesive bonds can be assessed if the cantilever has been previously functionalized or one cell is probed on top of another one; otherwise, nonspecific interactions are measured.

### 4.5. Assessing Cell Beating Force and Frequency

Cell beating characteristics which are of particular interest for studying ACM disease can also be measured using AFM. Here, the cells are gently touched by the cantilever tip (ideally a spherical tip) using, for example, 2 nN of force. The cantilever tip could be maintained in position for a minute interval while deflection data are collected. Deflection data can be transformed to force multiplying by the cantilever spring constant. The resulting data can provide force beating, frequency, duration (peaks distance), and full width at half maximum (FWHM) of each beat (Figure 5). One example of this technique in the literature is the work reported by Pena et al. [51] in which they evaluated the force beating of primary cardiomyocytes cultured in 3D hydrogels. They found that cells cultured on carbon nanotube-functionalized hydrogel had a more rhythmic beating with a stronger force.

Beating rate variability (BRV) could also be evaluated using the Poincaré plot (Figure 6). The Poincaré plot is a technique suggested by Tulppo et al. [52] for studying heart rate signals. It is a scatter graph of the peak’s interval plotted against the preceding peaks interval. The first peaks interval [53] denotes the x-coordinate, and the second interval [53] denotes y-coordinate. The quantitative analysis of the data is done by fitting an ellipse to the plot with its center corresponding with the centroid of the ellipse. The point where both ellipse axes meet relates to the total mean of the intervals. The length of minor and major axis of the ellipse are 2SD1, 2SD2, where SD1, SD2 are the dispersion perpendicular to the line of identity (minor axis) and along the line of identity (major axis), respectively, as previously described [52]. SD1 is the standard deviation of the instantaneous (short term) beat-to-beat variability while SD2 is the standard deviation of the long-term interval variability. Another index derived from this plot is the axes ratio R = SD1/SD2, measuring the balance between long- and short-term beating variability.

## 5. AFM for Cardiac Tissue Analysis

AFM is a well described technique that can be used to detect mechanical changes between biological samples at the nanoscale [54]. As discussed above, it is commonly used to study proteins and cells; however, more recently, AFM studies have been focused on the biomechanic assessments of tissues and organisms and the association of these measurements with biological mechanical functions, such as fibrosis. Although, AFM is gaining popularity to assess mechanical properties of tissues, such as cardiac tissue, there are still some aspects regarding (i) AFM methodology, (ii) sample preparation, (iii) data analysis, and (iv) AFM probes selections that may cause differences in results between studies. For example, as previously mentioned, it has been demonstrated that physical and physiological cues can affect the mechanical properties of samples. Indeed, the geometry and morphology of the AFM cantilever tip, the indenting force, the AFM mode, and the AFM analysis methodology will provide significant influence in the mechanical properties of biological samples (e.g., sharp cantilever tips create a two-fold increase in the value of the effective Young’s modulus relative to that of the blunt tips) [55,56]. Additionally, it is important to mention that the AFM does not provide bulk mechanical properties and that the mechanical properties detected by the AFM are at the nanoscale and relative to the physical and physiological cues of the analysis. Therefore, although the AFM is a powerful precise tool to identify differences between samples, the physical and physiological cues must be kept constant during the analysis in order to minimize the effect of external influences.

Perhaps one of the main questions regarding cardiac tissue analysis by AFM, or tissue analysis in general, involves the way in how the samples are needed to be prepared for analysis. Several investigations use fresh tissues to assess mechanical properties with AFM. However, the use of fresh tissue has several problems, such as (i) irregular cut with thickness variability within samples, (ii) the limitation to analyze several samples in a short period of time, and (iii) the difficulty to analyze the same time points in a large batch of samples. Although the evolution of more sophisticated microtomes may allow for more precise fresh tissue cuts, it is always difficult to have homogenous cuts with the same thickness between tissues due to the biological water content present in fresh tissues. In addition, if a large batch of samples need to be analyzed, the cut can be compromised since it has to be performed fast for immediate analysis to avoid tissue degradation. As previously mentioned, differences in tissue thickness will results in mechanical values discrepancy. Moreover, fresh tissue is not ideal when a large batch of samples are needed to be analyzed under precise time points (e.g., five animals per group at day 3 under several different conditions) because ideally, the samples need to be analyzed the same day that they are collected to avoid tissue degradation. Therefore, having a large batch of samples will complicate the use of fresh tissues. Since the AFM does not provide bulk mechanical properties, and it is mainly used to detect differences in mechanical properties between groups, frozen unfixed tissues may be more preferable to be used for AFM analysis than fresh tissues. In addition, the use of frozen tissues in AFM has been broadly documented [57,58,59,60,61,62]. This is mainly due to the feasibility to access and process frozen tissues (e.g., frozen tissues are easier to section, allowing for a more homogeneous surface than fresh tissues and can be stored for long period of times without compromising their biological integrity). In addition, large batches of tissues can be analyzed with the same time point in different days (e.g., a large batch of animals under different conditions can be sacrificed and the tissues collected at the same time point and later analyzed by AFM). Therefore, frozen tissues are more feasible for AFM analysis. However, both fresh and frozen tissues have been accepted for AFM analysis and they have both been broadly reported in the literature [63,64,65]. Fixed tissues have also been analyzed by AFM [66]; Iwashita et al. compared the stiffness of fresh brain tissues with brains fixed with 3% glyoxal solutions and 4% Paraformaldehyde solution. They found that glyoxal fixation does not introduce stiffness to the tissue, and thus can be used to assess for differences in brain tissue stiffness. Although fixed tissues may be ideal to avoid tissue degradation during AFM analysis, to our knowledge, this technique has not been yet used in cardiac tissues.

### Tissue Scanning Methodologies

There are two general methodologies to assess the mechanical properties of tissues using AFM: (i) force spectroscopy which has been discussed above, and (ii) fast scanning (e.g., JPK quantitative imaging, Bruker FastScan, etc.). In fast scanning, a high-definition imaging and force map can be obtained. Several studies have reported this technique to obtain high quality imaging on cardiac cells; one example is the work of Yang et al. which investigated the mechanisms of transmembrane transport of a synthetic polypeptide (GCIP-27) in cardiac cells. By using AFM, authors found a significant increase in the surface roughness of neonatal rat cardiomyocytes, when administrated with GCIP-27 [67]. A variation of cell surface roughness was detected even in neonatal mouse cardiomyocytes treated with aldosterone, as they were found to be more irregular than controls [68]. Myocardial infarction (MI) was proven to dramatically change the surface properties of cardiomyocytes: indeed, the crest/hollow organization, typical of adult murine cardiomyocytes, was compromised after MI [69]. Subcellular components, specifically the cytoskeleton, were investigated by AFM in neonatal mouse cardiomyocytes exposed to lipopolysaccharide (LPS), which is known to cause sepsis and cardiac dysfunction. LPS treatment has been shown to reduce the cytoskeletal density and increase cytoskeletal volume, thus leading to the conclusion that LPS causes the reorganization of cardiomyocyte cytoskeleton [70]. For purposes of comparison, in this section we will provide information only on the JPK quantitative imaging (QI) mode in regards of tissue analysis. Table 1 shows a comparison between force spectroscopy and quantitative imaging modes. As previously mentioned, force spectroscopy performs force–distance measurements in which the lateral position is set at a fixed point and the cantilever moves vertically towards and away from the surface. As the tip is pushed against the sample, mechanical property values can be measured from indentation. A variation of force spectroscopy is force mapping, which is an extension of the force spectroscopy mode. Here, the probe tip moves in the same way as in force spectroscopy, but the measurements are performed over a drip, providing values displayed as force mapping.

An advantage of force spectroscopy is that the analysis can be relatively fast when compared with quantitate imaging scanning. For example, 50 indentations around a heart tissue can be measured in ~30 min with high quality force curves (by using a microscope in combination with the AFM it is possible to control where the cantilever tip will land thus avoiding tissue irregularities and increasing the quality of the force curves). One of the best probes for tissue analysis with force spectroscopy is the Bruker MLCT. However, it is important to remember that the probe tip is in the nanometer range, and therefore the measurements obtained by force spectroscopy will be only of a nanometer indentation. Several authors have tried to increase the area of analysis using larger spherical probes (10 to 15 µm), however the large spheres can introduce noise vibration during the analysis which affect the quality of the force curves; in addition, a cantilever with a large spherical probe does not solve the fact that only a non-representative tissue area is indented, and so more people are moving to use force mapping. With force mapping, a drip can be used to set several indentations in a determined tissue area (e.g., 64 indentations in a 100 × 100 µm area). The problem with force mapping is that (i) it is difficult to control where probe will land (e.g., irregular tissue) and so, several force curves may need to be discarded, (ii) the force map generated has a very low-quality resolution, (iii) although high resolution stiffness maps can be obtained with force spectroscopy, it will take several hours for the analysis to be performed. An alternative to overcome these limitations is to use fast tissue scanning techniques, such as JPK quantitative imaging and Bruker fast scan. In the JPK quantitative imaging (QI) a whole force curve is measured at every pixel of the selected sample region, generating more than 60,000 force curves per scan and thus providing a high-resolution stiffness mapping of the sample when compared with force mapping (Figure 7).

In QI, the probe motion and sample rate have a different algorithm than force spectroscopy, and thus, it generates a higher imaging velocity. In addition, the piezo height is determined at 80% of the setpoint force and the force is held constant while the probe is scanned laterally over the surface. The main advantages of QI are that a more detailed information of tissue stiffness is provided since over 60,000 force curves are generated per scan. Sharp probe tips are better for QI analysis (e.g., qp-BioAC from NanoandMore), as they can provide a smaller indentation and thus a higher quality stiffness topography map. In addition, this high-resolution scan allows the correlation of specific tissue sections with mechanical properties (e.g., cell or extracellular matrix (ECM) stiffness can be precisely detected and correlated with the sample topography or with tissue staining). The disadvantages of QI are (i) long timing for tissue analysis; one scan in a 10,000 μm^2^ tissue section can take ~22 min and a minimum of four scans per sample are required to cover more tissue area and thus, to obtain more specific information; (ii) since the analysis can take around 1 h and 30 min per sample, a protease inhibitor solution should be used to stabilize the sample and avoid degradation; (iii) the AFM equipment can overheat and only 3 to 4 samples can be analyzed in a day. Since both force spectroscopy and QI have their limitations, a combination of both techniques can be used to optimize the assessment of tissue mechanics; for example, force spectroscopy can be used to quickly detect mechanical differences between samples and QI can be used for more in-depth analysis of a particular area of interest or just to provide a more visual representation of the mechanical properties of a tissue.

As previously mentioned, the most common model used for force curve analysis is the Hertz model. This model can be applied for both QI and force spectroscopy. Thanks to the Sneddon extensions of the Hertz model, different probe tip geometries can be used. The data generated for QI are numerical and imaging values for which force spectroscopy is only numerical values (also imaging if force mapping is used). While force spectroscopy can provide values for a specific location in the tissue (for example stiffness in between cells), QI generates thousands of values that can only be isolated precisely using a Matlab segmentation coding or a JPK software function to measure specific point values. This can be very important if, for example, values and percentage of ECM deposition are required in a tissue analysis. However, the JPK software function does not isolate all the values at the same time, and so they need to be isolated manually one by one. In order to isolate specific sections of a tissue (e.g., ECM) more efficiently, we engineered a MATLAB approach to segment the ECM values from the global QI data. Figure 8 shows a representative image of the Matlab segmentation approach. This approach relies on two steps, (i) identifying the ECM, and (ii) isolating the value of the identified area. To identify the ECM, the stiffness map image generated by the JPK software is used. This image has both a stiffness map and a scale (Figure 8A). Since only the stiffness map is needed, the scale needs to be cropped out (Figure 8B). The green channel of the cropped image can be then isolated. This is followed by a threshold of the green channel. Using the threshold, a binary mask can be created (Figure 8C). To isolate the values of the identified area a TSV file containing a list of (x,y) coordinates and Young’s modulus is needed. The list of position values can be used to create a mesh grid that mirrors the format of the image. The Young’s modulus values are then translated onto the grid. This creates a contour plot with a similar image to the JPK generated image (Figure 8D) The previously created binary mask can then be used to isolate the regions of interest in the contour plot. This newly generated plot displays just the regions of interest, and the user can use the cursor to hover over certain regions to get the values of an area of interest (e.g., ECM). Alternatively, the user can also generate a list of values of the region of interest.

Although the AFM only provides information regarding difference in tissue stiffness, it can be a complementary technique for other biological measurements; for example, it is well known that ECM influences mechanical tension in tissues [71,72], and thus AFM can be used as a powerful characterization tool to complement mechanistic changes in ECM remodeling [71] with techniques such as mass spectrometry, tissue staining, or second harmonic generation.

## 6. AFM as a Complementary Tool to Detect Fibrosis in Cardiac Tissue

ECM in the cardiac tissue: The main components of ECM are structural proteins, such as collagen and elastin, and non-structural proteins, such as proteoglycans, proteases, and growth factors [73,74,75]. From a mechanical standpoint, the most important of these proteins is collagen which is primarily found as collagen I and collagen III [76]. Other collagen types, such as collagen IV, V, and VI, are also present in the ECM [73]. The primary purpose of collagen, in the healthy myocardium, is to prevent overstretch, provide surfaces for myocytes to attach, and help in transmitting contractile forces to the cardiac tissue [77]. Other important ECM proteins are Fibronectin, which is involved in regulating the assembly of collagen I, an elastin which forms long and thin elastic fibers surrounded by microfibrils and proteoglycans which regulate water content [76].

The mechanical cues of the ECM act as a driver for a number of cell functions, including differentiation, motility, fibroblast activation, and collagen production [78]. Mechanical properties of the ECM can change drastically during physiological and pathophysiological conditions [78]. Under physiological conditions, the structural stability of the ECM is thought to offer, to the cardiac cell population, protection from any substantial change due to the mechanical stresses involved in the beating heart. Under pathological conditions, ECM plays an important role in cardiac remodeling; for instance, after a cardiac injury, a stable scar must quickly form to prevent ventricular wall rupture [79]. However, this process leads to cardiac fibrosis, which involves pathological myocardial remodeling characterized by cardiac fibroblast activation and differentiation into myofibroblasts, excessive deposition of ECM proteins, particularly type I and II fibrillar collagens. The over-production of ECM increases tissue stiffness, affects the physiological mechanics of the heart, results in impairment of cardiac contraction and relaxation, and leads to cardiomyocyte (CM) loss and progression towards heart failure.

Cardiac fibrosis can be classified in three categories: (i) reactive interstitial fibrosis, which is characterized by increased ECM deposition without a significant loss of (CM), (ii) infiltrative interstitial fibrosis, which involves glycolipid build up in different cells of the heart, and (iii) replacement fibrosis, which happens after a cardiac injury (e.g., myocardial infarction) where dead cells are replaced by a predominantly collagen type I scar [80].

Although collagen type I and II are the predominant proteins in the fibrotic heart [81], other ECM proteins are also present in the fibrotic heart, such as (i) fibrin and fibronectin which promote provisional plasma derived matrix (once lysed it forms an organized cell-derived second provision matrix of fibronectin and hyaluronan, leading to fibroblast migration and myofibroblast differentiation); (ii) Matrix-cellular proteins, which are a family of structural unrelated extracellular macromolecules and function as molecular bridges between matrix proteins and cells; (iii) collagen VI, which Kong et al. demonstrated that promotes myofibroblast differentiation. Therefore, there are a large variety of biomolecules involved in cardiac fibrosis [82].

Since fibrillar collagens are the most largely present in cardiac injury [83], the traditional methods to assess fibrosis only focus on detecting collagens I, II, and III. For this, two popular traditional methods are picrosirius red and second harmonic generation (SHG). Picrosirius red is one of the best understood techniques of collagen histochemistry, however this technique only visualizes collagen I and III fibers [84]. SHG is a more sophisticated imaging technique in which two photons with the same frequency interact with a nonlinear material are “combined” and generate a new photon with twice the energy of the initial photons (equivalently, twice the frequency and half the wavelength), providing a non-stained collagen imagen [85]. However, SHG cannot detect non-fibrous or symmetric fibrous collagen, and thus, collagen I and II are the strongest SHG signals and collagen III, although fibrous, results in a very week SHG signal [79]. Since there are many other ECM proteins and molecules involved in cardiac fibrosis, more sensitive methods should also be used to assess fibrosis. In this regard, AFM is a high-sensitive method to detect mechanical changes at the nanoscale caused by either matrix deposition (all types of collagens and other ECM proteins) and other molecules (e.g., glycolipids accumulation), and thus it can be used as a complementary high-sensitive tool to assess fibrosis in the heart.

Using a mouse myocardial infarction (MI) injury model, we performed a study in which we compared AFM with other popular techniques used to assess fibrosis: picrosirius red and SHG. Briefly, in C57BL/6 2-month-old mice, a left thoracotomy was performed via the fourth intercostal space. The left anterior descending coronary artery was ligated with an 8-0 silk suture near its origin between the pulmonary outflow tract and the edge of the left atrium for 45 min turning the anterior wall of the left ventricle (LV) pale. After the MI was induced, the ligation was released, the intercostal space, muscles and skin were sutured, and the mice were extubated to re-establish normal breathing. All anatomical structures were visualized with a stereomicroscope (Leica, Richmond, IL, USA). The procedures were performed according to the University of Colorado Denver Animal Care and Use Committee guidelines (COMIRB protocol 0079). Eight weeks post-MI, mice were sacrificed by cervical dislocation, the chest was cut open, the hearts were harvested, and cardiac perfusion with PBS was done in order to flush remaining blood. The hearts were then cut in two halves, as shown in Figure 9: (i) one section for the MI isolation which contained the MI region (close region) and (ii) one section for the rest of the heart tissue which was located in the far MI region (far region). The tissues were then embedded in optimal cutting temperature (OCT) compound, cryogenically cut with a 10 µm thickness, and analyzed by picrosirius red, SHG, and AFM. A brief methodology for these techniques is next described. For picrosirius red staining, frozen heart tissues sections were processed according to manufacturing protocol. Data was analyzed using a custom script written using Matlab R2018B in which the PSR script used thresholding of the green color channel, the HSV color space, and comparing intensities in the RBG color space to isolate the desired segments of the image. The green color channels were isolated from the image. The image was also transformed to the HSV color space. A threshold function was applied to the green color channel. A binary mask was created to meet the following conditions: (i) the red intensity is higher than both green and blue intensities at a pixel location, (ii) the green pixel intensity is lower than the threshold, (iii) the hue value in the HSV image was between 0.8 and 0, and (iv) the saturation value in the HSV image was between 0.125 and 1. Pixels meeting the conditions of the binary mask were isolated. The percentage of isolated pixels was calculated and printed to an excel sheet for further statistical analysis (GraphPad Prism 9). Using frozen heart tissues sections, autofluorescence and SHG signals were acquired using Zeiss 780 laser-scanning confocal/multiphoton-excitation fluorescence microscope with a 34-channel GaAsP QUASAR detection unit and non-descanned detectors for two-photon fluorescence (Zeiss, Thornwood, NY, USA). The imaging settings were initially defined empirically to maximize the signal-to-noise ratio and to avoid saturation, and they were kept constant for all measurements for comparative imaging and results. Two-photon Chameleon laser tuned to 800nm was used for excitation and emission signals corresponding to the autofluorescence and SHG signals were detected simultaneously through non-de-scanned detectors. Data was analyzed using a custom script written using Matlab R2018B in which the script started by isolating the red color channel from the image. A threshold function was used to set a threshold of pixel intensity. Pixels above the threshold were isolated. The percent of isolated pixels was calculated and printed to an excel sheet for further statistical analysis (GraphPad Prism 9). The methodology used for the AFM was based, and further adjusted to our own AFM equipment, on previous publications in which frozen biological tissues were used for mapping (4–7). Briefly, the mechanical properties of the cardiac tissues were analyzed using a NanoWizard^®^ 4a (JPK Instruments, Carpinteria, CA, USA). Frozen heart sections were allowed to thaw in PBS for 15 min to remove the OCT compound and remove any debris from the tissue. Tissue stiffness was determined using the QI mode with a qp-BioAC-1 (NanoandMore, Watsonville, CA, USA) cantilever with a force constant in the range of 0.15 to 0.55 N/m. Calibration of the cantilever was performed using the thermal oscillation method prior to each experiment. A 10,000 µm^2^ area was scanned using a set point of 5 nN, a Z-length of 2 mm. During all the analysis, the tissues were kept in a protease inhibitor solution (HALT 100% at a 1× concentration). All the physical and physiological cues regarding AFM analysis and sample preparation were kept constant in all our samples. Several cell orientations were scanned across the heart tissue, and they were considered for the mechanical average of the sample. Four random scans, including the MI scar, were performed per section. Every scan was composed with over 60,000 force curves (60,000 nanomechanical data points per scan). Vessels and heart boundary were not analyzed. The Hertz model was used to determine the mechanical properties of the tissues using the JPK software and a correction for an offset in the height data was performed line by line using the JPK data processing operation. The experimental data were analyzed using GraphPad Prism software using one-way ANOVA with Holm–Sidak’s multiple comparisons test for normal distributions or the Kruskal–Wallis test with a Dunn’s multiple comparisons test. Data in the text are reported as mean (normal distribution) values ± standard error.

Figure 9 shows the comparison between the techniques. As shown in this figure, both picrosirius red and SHG demonstrated that the close region of the MI samples has a higher amount of collagen deposition when compared with the far region of the MI samples, the close region of the sham samples, and the far region of the sham samples. However, both picrosirius red and SHG did not detect any significant differences in collagen deposition between the far region of the MI samples and the close and far region of the sham samples. When using AFM to assess for mechanical stiffness differences between samples, the far region of the MI samples was significantly different when compared with the close and far region of the sham samples. These results were not observed in the SHG or in the picrosirius red analysis. The results indicated that both SHG and picrosirius red are limited techniques to assess fibrosis since they are only able to detect changes in collagen type I, II and III. However, as mentioned, there are several other ECM proteins and molecules involved in cardiac fibrosis, and so more sensitive techniques need to be included to assess fibrosis. Although AFM is not able to elucidate which molecules are responsible for the increase in tissue stiffness, it can be a powerful tool to detect hidden fibrosis. For example, Travers et al. [61] found that in a model of diastolic disfunction (DD), cardiac fibrosis was not evident in any mouse cohort on the basis of picrosirius red staining or SHG microscopy. However, using AFM, they were able to find significant differences in tissue stiffness between the model of DD and the sham samples. They further performed mass spectroscopy which revealed induction in the expression of >100 extracellular matrix proteins in the model of DD when compared with the sham samples. Therefore, AFM can be used as a complementary highly sensitive technique to assess fibrosis in the heart, in particular hidden fibrosis.

## 7. AFM and Arrhythmogenic Cardiomyopathy

As mentioned above, AFM can be used to evaluate both morphology and mechanical properties in single cells and tissues. Here we present a few examples of how the AFM can be used to gather information regarding ACM pathology. In the case of ACM disease, it is well-established that mutations in genes coding for desmosomal proteins account for about 50% of the ACM cases, mostly in the adult population [86,87]. The pathways affected by mutations in genes coding for desmosomal proteins have been previously discussed in the review of Gao et al. and they are shown in Figure 10 [88].

As shown in Figure 10, the molecular pathogenical signaling pathways attributed to the development of the ACM phenotype inlcude: Hippo pathway activation and Wnt inhibition which enhanced adipogenesis, and the canonical TGFβ pathway which contributes to increased fibrosis in ACM through activation of SMADs and JNK. In addition, mutations in desmosome genes also destabilize the intercalated disks, causing dysfunction of ion channels with consequent electrical instability [88].

Desmosomal protein are: (i) cadherins (ii) armadillo proteins, and (iii) plakins. The cadherins desmogleins (DSGs) and desmocollins (DSCs) are transmembrane proteins, providing both the desmosome adhesive behavior and, with their tails, a bond to the armadillo proteins. The armadillo proteins, bind to desmoplakin (DSP) which, in turn, links the desmosome to the cytoskeleton desmin network. In the case of ACM, among desmosomal genes, PKP2, encoding for the protein Plakophilin-2 (PKP2), is one of the most commonly affected. One of the most widely used cell line to study ACM is the mouse atrial CM HL-1 cell-line. Overexpression of mutant PKP2 in HL-1 cells and single-cell force spectroscopy by AFM revealed that mutated PKP2 failed to interact with desmogleins DSP [89]. PKP-2 knock-down was sufficient to induce changes in cytoskeleton organization with perturbation of the actin network and changes in focal adhesions (AFM used for morphology analysis); as a consequence, stiffness decreased, alongside with a reduction in cell adhesion, suggesting an impact on CM-ECM interactions, supporting the concept that mechanotransduction is impaired in ACM [46]. In this study, the authors identified miR200b as one of the mediators of these effects, since miR200b predicted targets belonging to the focal adhesion pathways. In line with this, downregulation of miR200b partially rescued the mechanical properties of PKP2-deficient cells, with restoration of cellular stiffness but only partial actin network rescue, indicating that likely additional factors regulate the cytoskeleton organization.

HL-1 cells were also used to demonstrate that some DSG’s mutations disturb cell adhesion [90]. The authors found a decreased adhesion, but interestingly, DSG’s mutations seem to be more significant for cardiomyocyte adhesion than N-Cadherin. In another paper, overexpression of mutated LMNA in HL-1 cells show a decreased nucleus stiffness, again suggesting the idea that one of the key factors in ACM is a decreased cardiomyocytes resistance to stress [91].

The current literature supports the hypothesis that cells with mutations in the desmosomal proteins have an impaired adhesion behavior. This prompted the study the mechanical properties as well as the adhesion behavior of single proteins using the AFM for single molecule force spectroscopy (SMFS). Studies were dedicated to measure the properties of cell membrane proteins [92,93]. Furthermore, variants in the gene encoding the sarcomeric protein titin (TTN) were identified and correlated to ACM [94]. Titin is a protein which bridges the sarcomere along its longitudinal axis, interrelating end-to-end the Z disc and M band, respectively. Titin, a giant multi-domain protein, shows a spring-like behavior when helping the passive and restorative forces during sarcomere lengthening and/or shortening and is responsible for a large fraction of the diastolic force of cardiac muscle. Titin stiffness could be tuned by changing the expression ratio of the longer, more compliant N2BA-titin isoform, relative to the shorter, stiffer N2B-titin isoform. For this reason, several papers were also focused on measuring the titin mechanical characteristic using AFM-SMFS [95,96].

Even though mutations in desmosomal proteins are the ACM predominant cause, recently mutations in lamin A/C gene (LMNA), a type of intermediate filaments which form the major structural components of the nuclear envelope, have been discovered to be involved in a wide spectrum of laminopathies including ACM. AFM was used to characterize three lamin mutations, namely E161K, D192G, and N195K. In this case, AFM was used to assess cell stiffness, adhesion, viscoelasticity, and beating behavior (force frequency etc.) [45,97,98,99,100]. As far as the beating behavior is concerned, fundamental works on assessing the cardiomyocyte’s beating behavior is due to Liu et al. [101], Sun et al. [102], and Chang et al. [103]. Even though none of the abovementioned papers dealt with ACM pathologies, all of them used cardiomyocytes with cardiomyopathies and paved the way for the AFM use for assessing beating behavior in ACM. For example, the beating behavior for lamin mutations and the relationship between beating force and frequency, delocalization of Connexing 43, and citoskeleton microtubule network are described by Borin et al. [99]. Since life-threatening arrhythmias are one of the main clinical characteristics of ACM, AFM is a potential tool to study the beating behavior of ACM, using cardiomyocytes derived from human induced pluripotent cells (hiPSC-CM) from ACM patients.

Moreover, as previously mentioned, ACM affects both right and left ventricles, leading to arrhythmogenic right ventricular cardiomyopathy (ARVC) and arrhythmogenic left ventricular cardiomyopathy (ALVC) which are both inherited heart muscle diseases characterized by the progressive degeneration of cardiomyocytes, which are replaced by fibro-fatty tissue. In addition, fibrosis is also a pathological phenotype in ACM disease. Although the large majority of the studies using AFM to assess cardiac tissue are related to myocardial infarction and cardiac hypertrophy, Travers et al. [61] and our own original data demonstrated that AFM is a powerful tool to assess fibrosis in the heart that is not related to the traditional collagen type I and II deposition, and thus AFM can also be used as a highly sensitive technique to assess the biomechanics of ACM tissues.

## 8. Data Reliability: Variables to Be Considered

The AFM-based determination of the aforementioned variables is not an easy task due to various factors. They can be related to (i) the cantilever stiffness choice, calibration of its spring constant; (ii) the need to characterize the AFM tip before and after the measurements (ideally with electron microscopy); (iii) the experimental conditions such as place of cell testing, load speed, number of force curves recorded at the same place, and the type of stiff substrate where cells are plated; (iv) contamination or damage of the AFM probe tip during measurements as a source of artifacts; and (v) the data analysis (mainly the determination of the point of contact between the AFM tip and cell membrane), including the mechanical models applied to describe the cell mechanical behavior, etc. The other significant factors affecting the mechanics of cells are those influencing the cellular properties, such as culture conditions, the density of cell on the substrate, the number of passages, the day of testing, the temperature of the culture. For instance, only the change of the fetal bovine serum concentration from 10 to 5% decreases Young’s modulus for about 15–20%.

A simple modification of substrate surface where cells are plated, by coating it with poly-L-lysine, a common compound enhancing cell adhesion, influences the obtained data. In the case of tissues, the thickness and preparation of the sample will affect the stiffness of the tissue.

The need of high statistics demands a prolonged probing of every cell that can lead to remodeling of the actin network or even to damage of the cell membrane.

A separate group of factors influencing the obtained data is related to the theoretical model used for the quantification of the data obtained. In general, these model assumptions are only somewhat satisfied since they assume the cell as an isotropic, purely elastic material. For instance, there is still not sufficient information on frequency-dependent mechanical properties of living cells when dealing with cell viscoelasticity. One of the possible reasons is the lack of appropriate models that can be used to describe cellular elastic and viscoelastic properties based on AFM data.

All sources of possible errors/problems aforementioned seams highlight that the assessment of the absolute values of the cell mechanical characteristics is very difficult and may foster doubts in the worth of the AFM. However, first of all, the exact knowledge of the absolute values of the cell mechanical characteristics is not always needed. It can be overcome by comparing the results with reference cells but measured in the same experimental conditions. This requires a standardization of protocols that could be broadly applied, and that could supply a reference set of samples made independently in various laboratories.

## 9. Conclusions

ACM is an inherited heart muscle disorder which can cause sudden cardiac death, and thus, interest in new methodologies and approaches are growing to better understand the pathology of ACM. AFM is a great tool to assess the biomechanics of cardiac cells and tissues. Therefore, the main goal of the present work is to collect and review the most relevant investigations that used AFM to assess cardiac biological samples to highlight the potential of AFM as a tool to assess the biomechanics of ACM. Although none of the literature reviewed focused on ACM pathologies, there are several AFM methodologies, all of them discussed in this review, that can potentially be used to assess the biomechanics involved in ACM. From cardiac cell biophysics to tissue biomechanics, AFM is a unique tool to study biophysics of the heart under physiological and pathological conditions, and to assess the effects of therapeutic approaches at the cellular and tissue level. Although AFM does not provide specific information as to what is affecting the biomolecular properties of cells and tissues, it still is a great biomechanical tool to complement any biological assay, to better understand cardiomyopathies, and to develop therapeutics for cardiac pathologies.

## Figures and Tables

**Figure 1 ijms-23-03700-f001:**
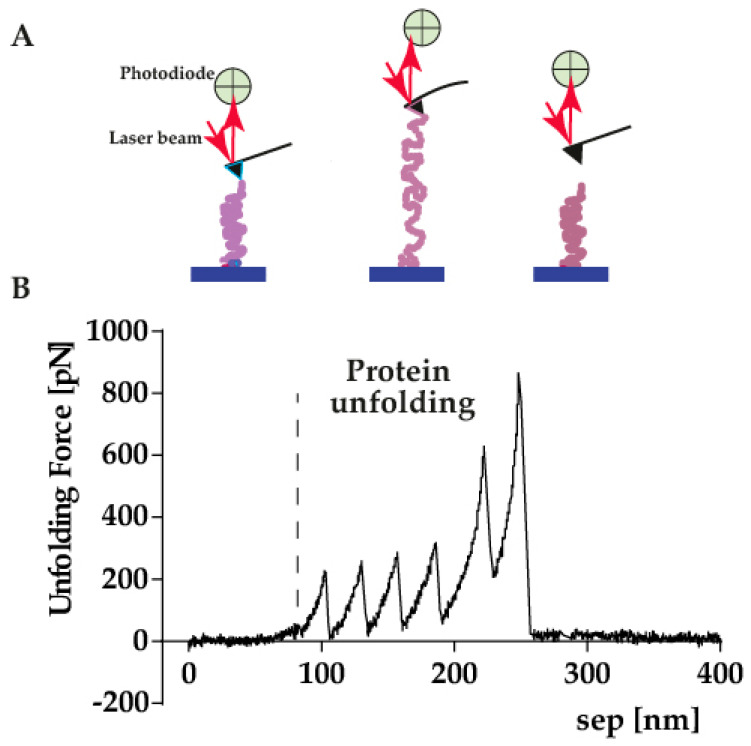
(**A**) Schematic representation of the AFM single molecule force spectroscopy experiment during unfolding of a protein. The AFM cantilever is lowered, and the pulling force is recorded during unfolding of the studied protein. (**B**) Force-extension curve showing a detailed spectrum of force peaks. Each force peak denotes an interaction that has been established by an unfolding intermediate. Fitting each force peak using the worm-like-chain (WLC) model reveals the contour length of the unfolded and stretched protein.

**Figure 2 ijms-23-03700-f002:**
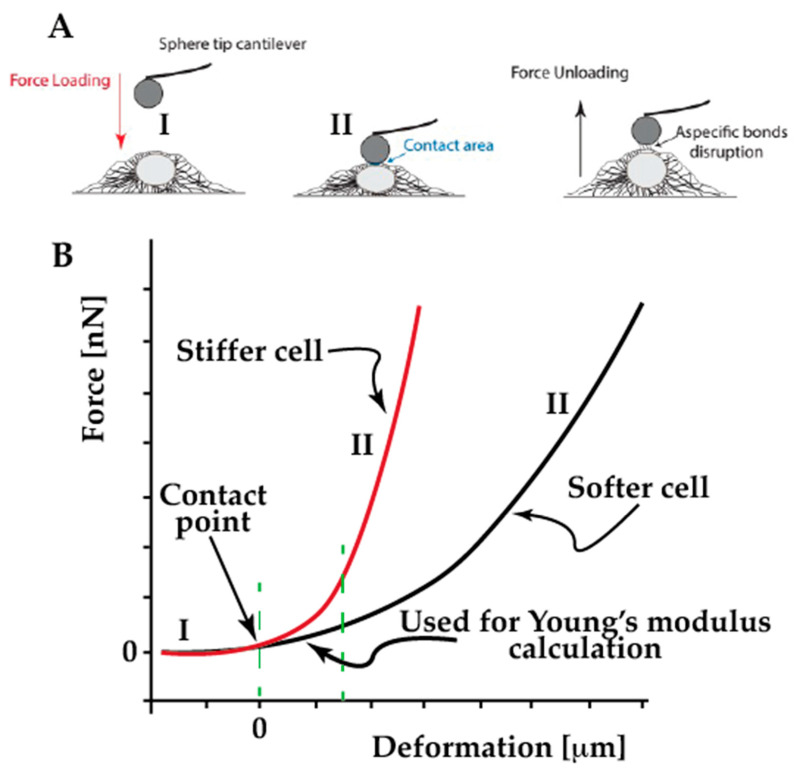
(**A**) Schematic representation of the AFM single cell force spectroscopy experiment. The AFM cantilever is lowered until it touches the cell, then the pulling force is recorded during loading and unloading. (**B**) Force–deformation curve during cell loading cycle, the cell (I-red) is stiffer than cell (II-black).

**Figure 3 ijms-23-03700-f003:**
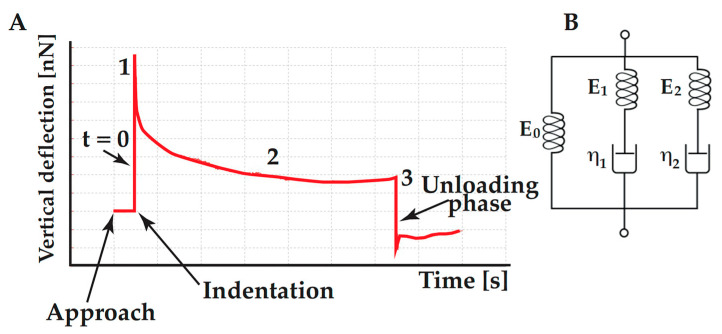
(**A**) Cartoon illustrating the viscoelastic test, (**B**) Generalized Maxwell model used to extract data from the viscoelastic test.

**Figure 4 ijms-23-03700-f004:**
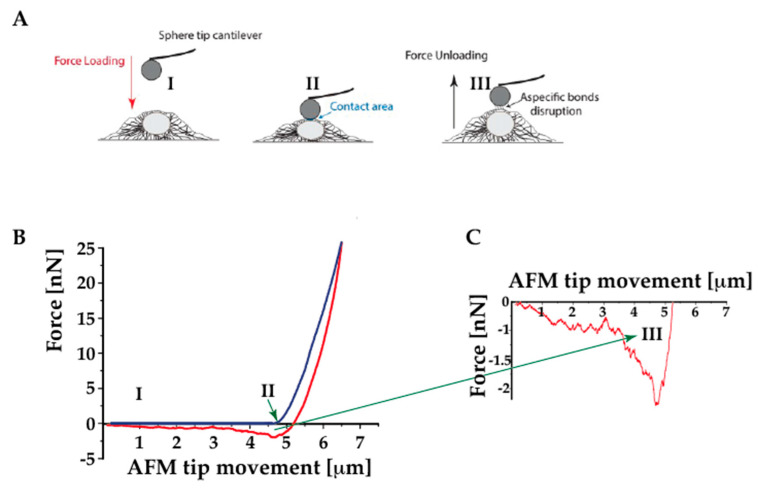
Introduction to the AFM force spectroscopy work for assessing adhesion behavior. (**A**) experimental setup of AFM force spectroscopy performed on the cell using a sphere tip; (**B**) sketch of the components of a F-D curve; (**C**) enlarged plot of the F-D retract part of the curve providing the adhesion data.

**Figure 5 ijms-23-03700-f005:**
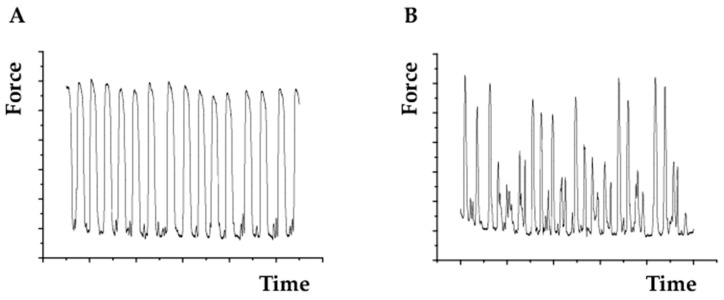
Example of beating recording for (**A**) a control cardiomyocyte cell and (**B**) a cardiomyocyte carrying the progeria mutation (c.1824 C > T, p.Gly608Gly).

**Figure 6 ijms-23-03700-f006:**
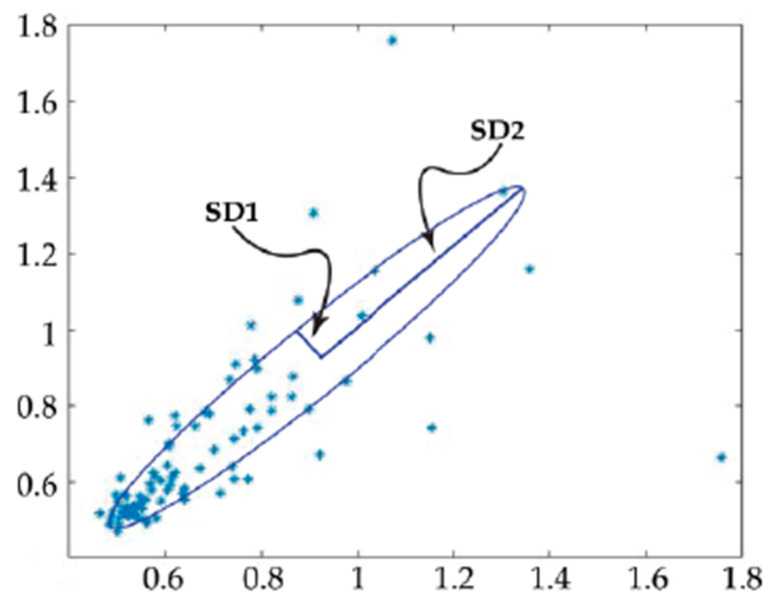
Example of a Poincaré plot obtained using the beating recording of the control cardiomyocytes in Figure 5.

**Figure 7 ijms-23-03700-f007:**
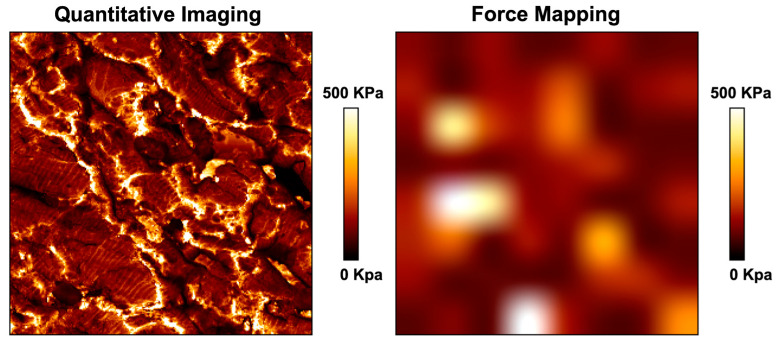
Stiffness mapping obtained by QI (**left**) and force mapping (**right**). Area analyzed 10,000 μm^2^.

**Figure 8 ijms-23-03700-f008:**
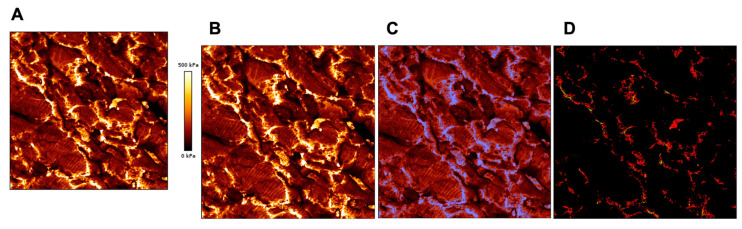
Representative image of the Matlab segmentation approach. Panels from left to right show: (**A**) original JPK image, (**B**) cropped image to removed scale, (**C**) cropped image with binary mask overlayed (blue) and (**D**) contour plot of isolated region.

**Figure 9 ijms-23-03700-f009:**
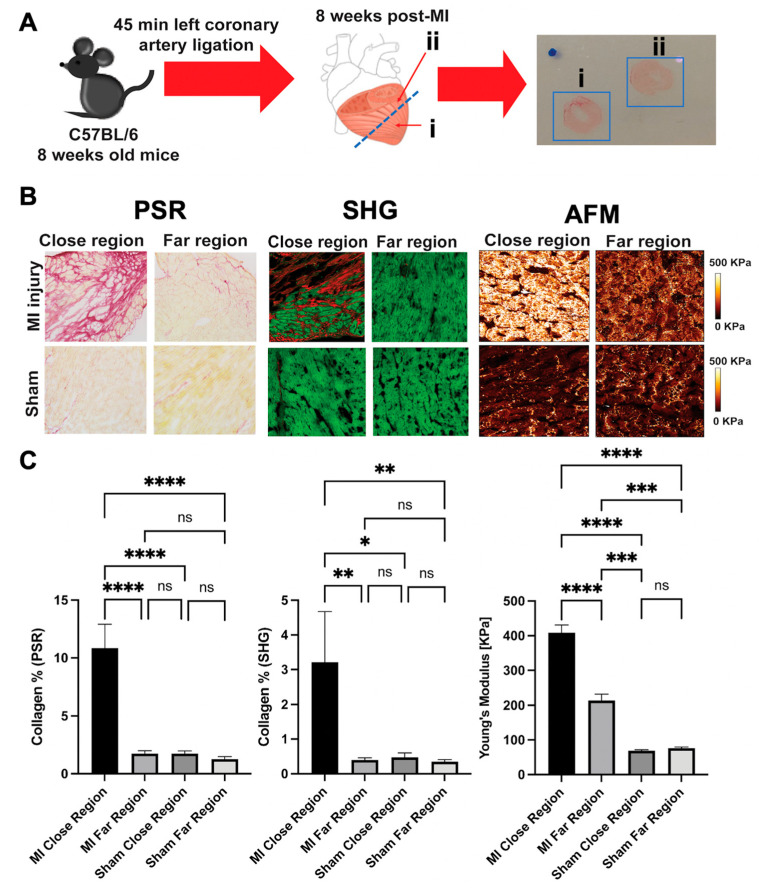
Comparison between Picrosirius Red staining (PSR), SHG, and AFM as techniques to detect fibrosis. (**A**) representative diagram of the sample collection: Eight weeks post-MI, hearts were harvested, cut in two halves, and cryogenically sectioned. Two sections on the hearts were analyzed. (i) One section was located close to the MI region (close region) and (ii) one section was located in the far MI region (far region). (**B**) Representative images of the hearts analyzed by Picrosirius Red staining, SHG and AFM techniques. (**C**) Quantification of collagen by Picrosirius Red staining and SHG and quantification of tissue stiffness by AFM: no significant differences in collagen deposition were observed between the MI far region and the sham (close and far regions); however, significant differences on tissue stiffness (Young’s modulus), analyzed by AFM, were observed between the MI far region and the sham (close and far regions). PSR: MI close region vs. MI far region **** *p* < 0.0001; MI close region vs. sham close region **** *p* < 0.0001; MI close region vs. sham far region **** *p* < 0.0001. ns: non-significant. SHG: MI close region vs. MI far region ** *p*: 0.0096; MI close region vs. sham close region * *p*: 0.0126; MI close region vs. sham far region ** *p*: 0.0082. AFM: MI close region vs. MI far region **** *p* < 0.0001; MI close region vs. sham close region **** *p* < 0.0001; MI close region vs. sham far region **** *p* < 0.0001; MI far region vs. sham close region *** *p*: 0.0005; MI far region vs. sham far region *** *p*: 0.0008.

**Figure 10 ijms-23-03700-f010:**
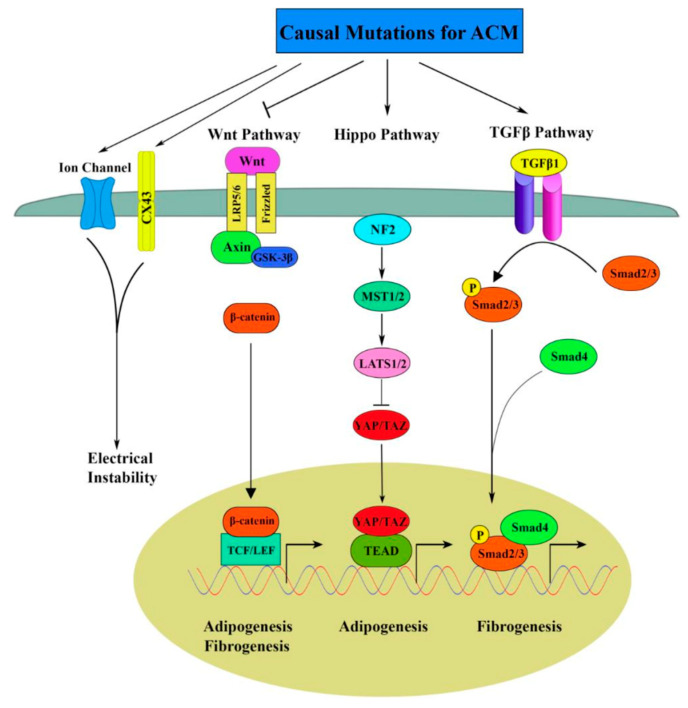
Molecular pathogenic signaling pathways in ACM. Reproduced with permission from Gao et al., International Journal of Molecular Sciences; Published by MDPI open access 2020 [88].

**Table 1 ijms-23-03700-t001:** General comparison between fast scanning and force spectroscopy methodology.

Quantitative Imaging	Force Spectroscopy
Advantages Provides tissue stiffness with topography mapping (imaging) It detects a more detailed and representative mechanical properties of the tissues (at the nanoscale) Disadvantages Long timing analysis Small area of scan: max 100 × 100 µm Analysis of the force curves is not very accurate; one algorithm fit should be applied for the 60,000 force curves generated in one scan	Advantages Fast AFM analysis, Force curves can be analyzed manually and so the algorithm fit is very accurate Disadvantages Data obtained may not be representative; even if doing 100 indentations across the sample, usually probe tips are too small (nanometers) to be comparable with fast scanning. The force mapping does not provide high quality force curves Not high-resolution mapping (force mapping) or not mapping at all for the force spectroscopy Not high-resolution topography or not topography at all for the force spectroscopy

## Data Availability

Not applicable.

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
