# Peer review of "Atomic Force Microscopy (AFM) Applications in Arrhythmogenic Cardiomyopathy"

_ijms, 2022, doi:10.3390/ijms23073700_

Round 1
Reviewer 1 Report
The Review entitled Atomic Force Microscopy (AFM) applications in arrhythmogenic cardiomyopathy is interesting and well structured. The atomic force microscopy technique is spectacular, so in order to understand the beauty of it, I strongly recommend adding at each addressed topic, figures, including representative examples, with the necessary permissions added.
Author Response
Reviewer 1
The Review entitled Atomic Force Microscopy (AFM) applications in arrhythmogenic cardiomyopathy is interesting and well structured. The atomic force microscopy technique is spectacular, so in order to understand the beauty of it, I strongly recommend adding at each addressed topic, figures, including representative examples, with the necessary permissions added.
We apologize if the figures were not included in the first version of the review most probably was due to technical issues uploading the file. Every “chapter” has al least one figure illustrating the technique and the results obtainable. We now included the Figures embedded in the text and not in a separated file to avoid this problem.
Reviewer 2 Report
The manuscript by Brisa Peña, Mostafa Adbel-Hafiz, Maria Cavasin, Luisa Mestroni and Orfeo Sbaizero reviews the use of AFM to applications in arrhythmogenic cardiomyopathy. The manuscript is very well written, comprehensive and up to date. The authors provided a very good introduction on the AFM technique and its force spectroscopy modes. Further on, a very extensive review of the most relevant investigations that used AFM to assess cardiac biological samples is provided. Review of AFM cardiac tissue analysis, AFM fibrosis detection in cardiac tissue and application of AFM in analysis of arrhythmogenic cardiomyopathy is presented. The value of this paper is further increased by practical remarks concerning AFM technique.
The paper is certainly suitable for publication and will be of interest to wide audience of biophysics and life science researchers alike.
Author Response
Thank you for the very nice and kind review
Reviewer 3 Report
The review paper is written in good English (but there are some minor typos, e.g. "proves" instead "probes", line 440) and contains 73 references. Atomic force microscopy based examination of cellular mechanical properties is broadly used in many areas and arrhythmogenic cardiomyopathy studies are a pretty narrow segment of this research field. There are plenty of high-quality reviews and even handbooks about the technique itself and it is not necessary to provide a detailed description of the method. Thus, a part of the introduction (lines 136-369) can be dramatically reduced. Indeed, the idea of atomic force spectroscopy is clearly presented even in some classic books (e.g. "Force-distance curves by atomic force microscopy", B. Cappella, G. Dietler, 1999) and in many modern handbooks, it is simply does not make sense to repeat it in a review paper about one pretty narrow application of this method.
It would be better to focus on arrhythmogenic cardiomyopathy itself, providing more information about the topic, including schematic illustrations of cardiac cells, typical design of experiment, etc.
Some statements should be made more accurate. E.g. in lines 441-443 the authors said: "However, it is important to remember that the AFM probes are in the nanometer range and so, the measurements obtained will be only of a nanometer indentation." A reader, who is familiar with AFM understands that they mean the probe's tip geometry (typically from 5 to 10 nm curvature radius), but not the probe itself (typically, a pyramid from 10 to 30 um height, mounted on the cantilever). It can be confusing for readers (e.g. cardiologists, who want to use a new method in their studies) who are not familiar with AFM.
Figures are mentioned in the manuscript, but they are not presented in the PDF file. Perhaps, there were some technical problems during the paper submission.
Author Response
Reviewer 3
The review paper is written in good English (but there are some minor typos, e.g. "proves" instead "probes", line 440) and contains 73 references.
We corrected all the “proves” typos and verified the grammar once again.
Atomic force microscopy-based examination of cellular mechanical properties is broadly used in many areas and arrhythmogenic cardiomyopathy studies are a pretty narrow segment of this research field. There are plenty of high-quality reviews and even handbooks about the technique itself and it is not necessary to provide a detailed description of the method. Thus, a part of the introduction (lines 136-369) can be dramatically reduced. Indeed, the idea of atomic force spectroscopy is clearly presented even in some classic books (e.g. "Force-distance curves by atomic force microscopy", B. Cappella, G. Dietler, 1999) and in many modern handbooks, it is simply does not make sense to repeat it in a review paper about one pretty narrow application of this method.
We reduced the sections from the previous lines 136 -369 (now 204 -783; we have more lines because the figures are now embedded in the text and few examples of the techniques were also included in the text). However, we kept some basic background because we believe that the readers (probably mainly medical doctors or clinical researchers) will appreciate having both basic background and examples in how to apply the techniques in one file.
It would be better to focus on arrhythmogenic cardiomyopathy itself, providing more information about the topic, including schematic illustrations of cardiac cells, typical design of experiment, etc.
We included one more Figure (Figure 10 in page 20) that shows the mechanistic pathways that induce ACM. We also included more examples of AFM techniques applied in cardiac cells.
Some statements should be made more accurate. E.g. in lines 441-443 the authors said: "However, it is important to remember that the AFM probes are in the nanometer range and so, the measurements obtained will be only of a nanometer indentation." A reader, who is familiar with AFM understands that they mean the probe's tip geometry (typically from 5 to 10 nm curvature radius), but not the probe itself (typically, a pyramid from 10 to 30 um height, mounted on the cantilever). It can be confusing for readers (e.g. cardiologists, who want to use a new method in their studies) who are not familiar with AFM.
We appreciate the feedback; we now explained better the lines 441- 443 (now lines 909 - 912) and other statements across the manuscript for more clarity.
Figures are mentioned in the manuscript, but they are not presented in the PDF file. Perhaps, there were some technical problems during the paper submission.
We apologize for this difficulty. Indeed, we had some technical problems uploading the figures and to avoid this problem, now we embedded the figures in the manuscript.
Round 2
Reviewer 3 Report
The manuscript has been improved and I have only a couple of minor comments to the authors:
- You highlight a very important topic (reproducibility of AFM based mechanical measurements, lines 82-94). I completely agree, but it would be better to add contamination or damage of the AFM probe tip during measurements as a source of artifacts and remind that it is necessary to characterize the tip before and after the measurements (ideally, with electron microscopy, but even just scanning of a sharp calibration grid can be enough).
- Figure 1. Why the illustration of the second step of unfolding contains a lateral deformation of the cantilever (laser comes to the left side of the detector)?
Author Response
Thank you for the commnets
Regarding Figure 1 you were perfectly right, the arrows were in the wrong position/direction. Now we slightly the Figure 1.
Regarding the possible problems during the AFM tests we change the statement as
....The AFM-based determination of the aforementioned variables is not an easy task due to various factors. They can be related to (i) the cantilever stiffness choice, calibration of its spring constant; (ii) the need to characterize the AFM tip before and after the measurements (ideally, with electron microscopy) ; (iii) the experimental conditions such as place of cell testing, load speed, number of force curves recorded at the same place and the type of stiff substrate where cells are plated; (iv) contamination or damage of the AFM probe tip during measurements as a source of artifacts; and (v) the data analysis (mainly the determination of the point of contact between the AFM tip and cell membrane), including also the mechanical models applied to describe the cell mechanical behavior, etc. We hope we fulfilled all the requests Best regards orfeo sbaizero